# Human γδ TCR Repertoires in Health and Disease

**DOI:** 10.3390/cells9040800

**Published:** 2020-03-26

**Authors:** Alina Suzann Fichtner, Sarina Ravens, Immo Prinz

**Affiliations:** 1Institute of Immunology, Hannover Medical School, 30625 Hannover, Germany; Ravens.Sarina@mh-hannover.de (S.R.); Prinz.Immo@mh-hannover.de (I.P.); 2Cluster of Excellence RESIST (EXC 2155), Hannover Medical School, 30625 Hannover, Germany

**Keywords:** γδ T cells, γδ TCR repertoires, TCR diversity, innate T cells

## Abstract

The T cell receptor (TCR) repertoires of γδ T cells are very different to those of αβ T cells. While the theoretical TCR repertoire diversity of γδ T cells is estimated to exceed the diversity of αβ T cells by far, γδ T cells are still understood as more invariant T cells that only use a limited set of γδ TCRs. Most of our current knowledge of human γδ T cell receptor diversity builds on specific monoclonal antibodies that discriminate between the two major subsets, namely Vδ2^+^ and Vδ1^+^ T cells. Of those two subsets, Vδ2^+^ T cells seem to better fit into a role of innate T cells with semi-invariant TCR usage, as compared to an adaptive-like biology of some Vδ1^+^ subsets. Yet, this distinction into innate-like Vδ2^+^ and adaptive-like Vδ1^+^ γδ T cells does not quite recapitulate the full diversity of γδ T cell subsets, ligands and interaction modes. Here, we review how the recent introduction of high-throughput TCR repertoire sequencing has boosted our knowledge of γδ T cell repertoire diversity beyond Vδ2^+^ and Vδ1^+^ T cells. We discuss the current understanding of clonal composition and the dynamics of human γδ TCR repertoires in health and disease.

## 1. Introduction

γδ T cells are detected at frequencies of 3–10% of T cells in the peripheral blood of human adults and are often enriched as resident cells within solid organs and mucosal tissues [1,2,3]. The biology of γδ T cells in blood and tissues is incompletely understood, although they exert pleiotropic functions such as cytokine production, tissue regulation, B cell help and cytotoxicity [4]. First, the defining characteristic of the γδ T lymphocyte subset is their specific T cell receptor (TCR), composed of a γ-chain (TRG) and a δ-chain (TRD). The genes encoding TRG and TRD rearrange during γδ T cell maturation in the thymus.

Briefly, the somatic DNA recombination of variable (V), diversity (D, only in TRD), and joining (J) elements creates combinatorial diversity of the individual TCR chains, a process called V(D)J-recombination [5]. Next to a multiplication of the potential TCR variety by the pairing of TRG and TRD chains, overall diversity is greatly amplified by junctional diversity, through the insertion of palindromic sequences (P nucleotides) and of non-templated nucleotides by the terminal deoxynucleotidyl transferase (TdT) enzyme (N nucleotides) at the V(D)J junction (CDR3 region) [6]. Therefore, all γδ T cells together possess a large repertoire of unique TCRs, termed clonotypes, that in theory could comprise up to 10^18^ TRG/TRD combinations [5]. However, the number of clonotypes found in an individual’s γδ TCR repertoire is probably much smaller, and the composition of human γδ TCR repertoires in health and disease is an active field of research.

For a long time, the general understanding of γδ T cell biology was that γδ T cells are innate-like T lymphocytes, similar to invariant natural killer T (NKT) or mucosa-associated invariant T (MAIT) cells. This was in part because, in contrast to αβ T cells that recognize peptide antigens in a conserved MHC-restricted mechanism, ligands and factors that shape the γδ TCR repertoire and activation remained largely enigmatic. Only a few direct TCR ligands were identified to date. Those are either endogenous MHC-related (e.g., endothelial protein C receptor (EPCR), MR1 or CD1d) or MHC-unrelated proteins (e.g., annexin A2) [7,8,9,10]. In particular, EPCR and annexin A2, as well as phosphorylated metabolites of isoprenoid synthesis, were described as serving as self-antigens that indicate cellular stress [5,11,12,13]. Most importantly, B7 receptor family-like butyrophilin (BTN) and butyrophilin-like (BTNL) molecules have been implied in the development of specific epithelial and circulating γδ T cell subsets [14,15,16,17] and as direct γδ TCRs ligands [18,19,20,21]. Advances in next-generation sequencing (NGS) analysis of human γδ TCR repertoires, together with the recent identification of γδ TCR ligands, shed light on the vast TCR diversity of human γδ T cells, thereby pointing to a complex role in health and disease. These studies support the idea that γδ T cells have features of innate and adaptive immune cells, that may depend on their developmental origin and priming, and hence may explain their multifaceted roles in tissue homeostasis, autoimmunity, pro- and anti-tumor activity, and during infectious diseases.

## 2. Human γδ T Cell Subsets Are Defined by Their TCR δ Chain

In mice, tissue localization and effector function of γδ T cells is typically correlating to their expressed TCR γ chain. For instance, (according to the Heilig and Tonegawa nomenclature [22]) skin-surveilling Vγ5^+^ T cells exclusively locate to the skin epidermis. In contrast, Vγ7^+^ T cells reside as specialized intraepithelial lymphocytes in the gut, and Vγ1^+^ T cells circulate as naïve or IFN-γ-committed T cells in the periphery. Furthermore, IL-17-producing Vγ4^+^ or Vγ6^+^ T cells are enriched in tissues such as the dermis, oral mucosa, brain, joints or reproductive tracts [23,24]. Similarly, human γδ T cells can be roughly grouped by V-gene usage (Table 1).

According to the international ImMunoGeneTics information system® (IMGT) [26], the human TRG locus is encoded on chromosome 7 and contains six functional V gene segments, called Vγ2 to Vγ5, Vγ8 and Vγ9 (TRGV2, TRGV3, TRGV4, TRGV5, TRGV8 and TRGV9), five J-elements (TRGJ1, J2, JP1, JP2 and JP) and two constant gene regions (TRGC1, TRGC2) [25,26,40,41]. The V-genes TRGV2-5 and TRGV8 have a relatively high sequence similarity and differ from the TRGV9 sequence. TCRs using the latter TRGV9 element are therefore sometimes still (misleadingly) called Vγ2^+^ instead of Vγ9^+^ [27].

The TRD locus is situated within the α-chain (TRA) locus on chromosome 14 and includes eight functional V gene segments called Vδ1 to Vδ8 (TRDV1, TRDV2, TRDV3, TRAV14/DV4, TRAV29/DV5, TRAV23/DV6, TRAV36/DV7 and TRAV38/DV8), with TRDV1-3 genes being used most frequently, along with three diversity (TRDD1-3), four joining (TRDJ1-4) and one constant (TRDC) gene region [26,40,41]. It is well established that most Vγ9^+^ chains assemble with the TRGJ element “JP” and often pair with Vδ2^+^ sequences. The resulting semi-invariant Vγ9(JP)Vδ2^+^ TCR is expressed by innate-like Vγ9Vδ2^+^ T cells [42,43,44,45]. Vγ9Vδ2^+^ T cells are considered the main circulating γδ T cell subset in humans [31,33,46]. They are the major γδ T cell fraction in the peripheral blood of most adults and expand after birth, likely upon the sensing of host- or microbe-derived prenyl pyrophosphates, also called phosphoantigens (pAgs) [11,12,44,47,48,49] (Figure 1). In brief, pAgs are metabolic products that interact with the butyrophilin family member BTN3A1 and activate Vγ9Vδ2 T cells in a BTN2A1-dependent manner [15,16,18,21,50,51,52]. BTN2A1 has been shown to interact with BTN3A1 and binds Vγ9^+^ chains via germline-encoded residues in the hypervariable region 4 (HV4) and CDR2, similar to BTNL3 interactions with Vγ4^+^ chains [18,19,21]. According to Rigau and colleagues, phosphoantigen reactivity depends on the Vγ9JP CDR3 loop and CDR2 residues of Vδ2 chains, that seem to form a second interaction site with another molecule (potentially BTN3A1) on the TCR surface [43,45,53]. Conversely, Karunakaran et al. confirmed the BTN2A1-CDR2δ interaction and also observed CDR3δ to be crucial for pAg reactivity, but proposed a composite ligand model involving Vγ9 germline-mediated BTN2A1 recognition and CDR3-dependent binding to one or more separate ligands [21]. Phosphoantigens are produced by several bacteria and stressed cells (e.g. virus-infected or transformed cells) and are potent antigens leading to the rapid anti-bacterial, anti-viral or anti-cancer responses of Vγ9Vδ2^+^ T cells [12,13,54]. These small molecules interact with an intracellular domain of BTN3A1 (B30.2) and are thought to induce conformational changes that lead to Vγ9Vδ2^+^ T cell activation [15,17,50].

Other human γδ T cell subsets are non-reactive to phosphoantigens, and include cells that use Vδ2 chains paired with non-Vγ9 chains, also known as Vγ9^–^Vδ2^+^ T cells, or display Vδ1^+^ or Vδ3^+^ γδ TCRs [31,33,34,42]. Such non-Vγ9Vδ2^+^ T cells can undergo clonal expansion and often represent the dominant γδ T cell fraction in tissues; albeit some adult individuals also display high frequencies in peripheral blood [33,34].

## 3. γδ T Cell Subsets Arise Early during Ontogeny

In mice, the development of several waves of γδ T cell subsets, such as Vγ5Vδ1^+^ dendritic epidermal T cells (DETCs) or Vγ6Vδ1^+^ IL-17-producing γδ T cells, exclusively takes place in the fetal thymus and these cells are maintained as long-lived effector cells after birth [23,24]. Early ontogenetic murine γδ T cell subsets are characterized by the expression of highly invariant and public TCRs that are shared among individual mice [55]. This publicity is presumably caused by simple gene rearrangements and/or positive selection during thymic development [56]. While Vγ6Vδ1^+^ T cells are prewired to become IL-17 producers even before TCR expression [57,58], a strong TCR-signal mediated via Skint1, a butyrophilin-like molecule expressed on thymic epithelial cells, will induce the IFN-γ phenotype of Vγ5Vδ1^+^ DETCs during thymic development. Later on, the murine thymus produces different γδ T cell subsets that often display a high TCR repertoire diversity.

Recent advances showed that, comparable to the mouse γδ T cell compartment, it is likely that human γδ T cells also arise in developmental waves. During fetal development, Vδ1^+^ TCR rearrangements dominate the fetal thymus γδ T cell receptor sequences (gestational week 15 and 16) [59]. Later in gestation, both Vδ1^+^ and Vδ2^+^ TCRs can be detected, with a prominence of Vδ2^+^ chains [59,60,61]. Along that line, the predominant γδ T cell subset in the fetal blood consists of Vγ9Vδ2^+^ T cells (75%–80%) in the second trimester and low frequencies of Vδ1^+^ and Vδ3^+^ γδ T cells (< 5%), indicating a first wave of Vγ9Vδ2^+^ T cells before gestational week 30 [62]. These fetal Vγ9Vδ2^+^ T cells are characterized by a semi-invariant Vγ9Vδ2^+^ TCR with characteristics of pAg-reactive TCRs, like TRGV9-TRGJP rearrangements and restricted CDR3γ lengths [43], and were shown to be reactive to the phosphoantigen (*E*)-4-Hydroxy-3-methyl-but-2-enyl pyrophosphate (HMBPP) [62]. Interestingly, fetal blood Vγ9Vδ2^+^ T cells show an enrichment of a public germline-encoded CDR3 nucleotide sequence “5’-TGTGCCTTGTGGGAGGTGCAAGAGTTGGGCAAAAAAATCAAGGTATTT-3’’ (translation: “CALWEVQELGKKIKVF”), formed without the addition of N nucleotides, but using short homology repeats (GCA) [28,62]. TdT expression in thymocytes increases with age and low levels were shown to favor the generation of such germline-encoded TCR clones (without N nucleotides) during early thymic development. Later on during ontogeny, the length of each CDR3 region is more variable, and a higher number of N insertions is used. Thus, in adults, this specific public germline-encoded clone is found in varying frequencies, and is mostly generated using N additions. This leads to a differential nucleotype usage of fetal and adult germline Vγ9JP clones [28,33,63]. Its CDR3 region seems to be prototypic for pAg recognition by Vγ9Vδ2^+^ T cells, and is a major contributor to the length homogenization to around 14 amino acids observed in the Vγ9 chain repertoire [62,64,65]. We might speculate that this simple rearrangement, and thus the length homogenization, coevolved with the pAg sensing ability of Vγ9Vδ2^+^ TCRs and the conserved binding of Vγ9 to BTN2A1 [18,21]. Thus, by retaining the binding capacity to BTN2A1, the Vγ9JP chain may pair with a multitude of Vδ2^+^ chains in the blood [66], creating a high TCR repertoire diversity of Vγ9Vδ2^+^ T cells. Further studies should address the effect of BTN2A1, and potentially other factors, to positively select Vγ9Vδ2^+^ T cells during thymic development.

The earliest T cells, namely the subset of pAg-specific Vγ9Vδ2^+^ T cells, are followed by Vδ1^+^ T cells that can be detected in fetal blood at week 25 and increase to become the major population of γδ T cells at term-delivery [37,62,67] (Figure 1). The γδ TCR repertoire of fetal non-Vγ9Vδ2^+^ thymocytes was shown to comprise an oligoclonal TRG repertoire, a diverse TRGV usage (including the non-functional TRGV10) and usage of mainly TRDV2 rearrangements paired with TRDJ2 or TRDJ3 [36]. Those fetal thymocytes use few N insertions, and TdT expression at this stage is low [36]. Similar to germline-encoded Vγ9JP rearrangements, invariant TCRs (TRGV8JP1, TRGV10JP1, TRDV2D3, TRDV1D3) were found to be expressed by fetal non-Vγ9Vδ2^+^ thymocytes, and recombination is thought to be similarly dictated by short homology repeats [36]. Using OP9DL1 cultures, Tieppo and colleagues showed the involvement of Lin28b in the induction of an effector program, the inhibition of TdT expression and the formation of germline-encoded CDR3γ and δ sequences in fetal thymocytes. The fact that fetal non-Vγ9Vδ2^+^ T cells (in particular one Vγ8Vδ1^+^ T cell clone) can already mount efficient immune responses against in utero cytomegalovirus (CMV) infections [37] supports the idea that γδ T cells are an important innate immune cell subset during fetal life and in neonates.

## 4. Development and Maintenance of γδ T Cells in Child- and Adulthood

In the postnatal thymus, Vδ1^+^ T cells are the most abundant γδ T cell population and Vδ2^+^ chains are found at very low levels [28,47,61,68,69]. The FACS monitoring of γδ T cell frequencies (Vδ1^+^ and Vδ2^+^) in peripheral blood lymphocytes of young children and pediatric thymi lead to the conclusion that γδ T cells undergo an extrathymic, postnatal maturation in response to environmental stimuli during early childhood [47]. Moreover, human immature γδ T cells (mainly Vδ1^+^) leave the postnatal thymus to differentiate into cytotoxic T cells in the periphery [68]. Recent NGS studies have supported this view, as human γδ TCR repertoires are highly polyclonal in the pediatric thymus and cord blood [33,34,69,70,71,72] and adult TCR repertoires appear less diverse and highly focused [33,34,71]. Thus, circulating fetal-derived Vγ9Vδ2^+^ T cells probably undergo a postnatal expansion, driven by the exposure to phosphoantigens of bacterial origin or food products after birth [47,54,73] (Figure 1). For Vγ9Vδ2^+^ T cells, TRD repertoires of fetal blood lymphocytes are characterized by shorter CDR3 lengths, as compared to adult Vδ2^+^ TRD repertoires, and the preferential usage of Vδ2 rearrangements with TRDJ3, and to a lesser extent TRDJ2 and TRDJ1 gene segments [28]. In contrast, adult Vδ2^+^ chains show a bias for TRDJ1 usage [28,29,33] and contain different nucleotides encoding the germline-derived public Vγ9JP clone that is present in every individual [31,33,63]. This led to speculation on whether postnatal selection or postnatal thymic output can explain this major change in the J-usage of human Vγ9Vδ2^+^ TCR repertoires [31,62]. Recently, Papadopoulou and colleagues tracked the lineage relationship of Vγ9Vδ2^+^ T cells and confirmed that the majority of adult blood Vγ9Vδ2^+^ T cells derive from a small subset of postnatal Vγ9Vδ2^+^ thymocytes that show adult-like features, e.g., TRDJ1 usage, and represents around 6% of postnatal thymocytes [28]. Interestingly, the phosphoantigen-dependent expansion of neonatal or adult Vγ9Vδ2^+^ T cells in in vitro assays did not affect TRDJ usage or diversity [28,30], and the proliferated cells retained a high oligoclonality. This suggests that adult Vγ9Vδ2^+^ TCR repertoires represent a blend of adult-like Vγ9Vδ2^+^ TCR clonotypes and a few remaining fetal-derived clonotypes that underwent postnatal expansion events. Nevertheless, it is tempting to speculate that recurrent pAg education might induce a slow, still polyclonal, outgrowth of some Vγ9Vδ2^+^ T cell clones, as observed in some adults [29].

The TCR repertoires of postnatal thymic non-Vγ9Vδ2^+^ T cells, mostly Vδ1^+^, have been reported to be extremely polyclonal [36,69,70]. In the thymus, postnatal Vδ1^+^ T cells use various TRGV gene segments, yet with a distinct preference for TRGJ1. In addition, Vδ1^+^ T cells use a high fraction of shared TRG sequences, whereas the corresponding TRD repertoires are largely non-overlapping, and were thus described as mostly private [69]. In contrast to fetal thymocytes, TdT expression is high in the postnatal thymus, and thus the usage of short homology repeats is inhibited, while the number of N additions used increases, leading to a repertoire distinct from fetal repertoires [36,74].

A key question of studying TCR repertoire composition is whether snapshots of repertoires are representative of a steady state, or how volatile γδ TCR repertoires are in a healthy individual’s life. We do not have longitudinal data from cord blood to adulthood, however, systematic comparison of adult peripheral blood versus cord blood-derived total γδ or Vδ1^+^ TCR repertoires points out that single clonotypes can expand from a diverse neonatal Vδ1^+^ T cell pool [33,34]. In healthy adult individuals, γδ TCR repertoires were shown to remain stable over a time of at least 90 days, indicating that changes in the γδ repertoire are most likely caused by more severe immunological challenges and do not usually happen at steady state [33]. The fact that some healthy adults showed no expanded single Vδ1^+^ clonotypes, further leads to the conclusion that postnatal Vδ1^+^ repertoire focusing is not generally caused by T cell maturation, but is more likely an effect of specific antigenic challenges [34]. In line with this, stable γδ TCR repertoires in chronic hepatitis virus C (HCV) patients and during direct-acting antiviral drug therapy were observed [35]. Intriguingly, the de novo generation of human γδ T cell repertoires from stem cells, after allogeneic hematopoietic stem cell transplantation (alloHSCT), led to a reconstitution of γδ TCR repertoires that showed comparable diversity and quality to repertoires of healthy adults [33]. Interestingly, the same study revealed that distinct new Vδ1^+^ and Vδ2^+^ clonotypes arose from donor stem cells, indicating *de novo* generation in the adult thymus. It will be interesting to investigate whether the same level of functionality of those newly generated γδ TCR repertoires is restored in alloHSCT recipients [33,72].

Upon ageing, Kallemeijn and colleagues reported shrinking of the naïve γδ T cell population (CD45RA^+^CD27^+^CD197^+^), while repertoire diversity was maintained [70,75]. Moreover, a tendency for a decreased Vγ9 usage and an increase of Vγ2-5 and Vγ8 chain usage in elderly individuals was shown for effector (CD45RA^–^CD45RO^+^CD27^–^CD197^–^) and central memory γδ T cells (CD45RA^–^CD45RO^+^CD27^+^CD197^+^) [70,75], as well as a general reduction of paired Vγ9Vδ2^+^ TCRs in some individuals [66]. Furthermore, γδ TCR repertoires in healthy elderly individuals are characterized by large clonal expansions of particular non-Vγ9Vδ2^+^ clonotypes that could reflect the history of antigen challenge [66].

## 5. Impact of Infectious Diseases on γδ TCR Repertoires

A role for γδ T cells in the course of viral, bacterial or parasitic infections has been proposed by many studies via flow cytometric assessment of γδ T cell quantities and/or qualities in the onset, progression and prognosis of infections [76,77,78]. Recent advances in the NGS analysis of peripheral human γδ TCR repertoires gave strong evidence that non-Vγ9Vδ2^+^ T cell subsets can mount an adaptive-like immune response [33,34]. Yet, data monitoring the TCR repertoire of γδ T cells during infections is still scarce and only a limited number of diseases have been studied.

### 5.1. Viral Infections

Vδ1^+^ T cells have long been associated with the immune response after human Cytomegalovirus (CMV) infection, as firstly described by the Déchanet-Merville group in the context of kidney transplantation [37,79,80]. More recent NGS TCR repertoire analyses added the important detail that CMV-driven expansions of non-Vγ9Vδ2^+^ T cells were indeed clonal, e.g., in patients with CMV reactivation after alloHSCT [33] and in CMV-seropositive healthy adults [31,34]. Furthermore, expanded CD8^+^ γδ T cells were found in CMV-positive grafts [81]. In stem cell transplant recipients, CMV re-activation induced an immediate clonal expansion of individual non-Vγ9Vδ2^+^ T cell clones that was still visible after viral clearance, indicating a memory formation of virus-induced γδ T cell clones [33]. In contrast, patients without CMV reactivation showed high γδ TCR repertoire stability in longitudinal samples. Similarly, CMV-positive adults have skewed Vδ1^+^ TCR repertoires [34]. In addition to Vδ1^+^ T cells, Vγ9^–^Vδ2^+^, a prevalent Vδ2^+^ subset at birth with diverse Vγ chain usage, also show clonal expansions and transition from a CD27^hi^ naïve-like to a CD27^lo/neg^ effector-like phenotype after acute CMV [31,32]. Together, these studies support the idea that individual expanded clones are unique to donors and most likely stem from the selection of low frequency clones that expand upon antigen challenge. As a consequence, in contrast to Vγ9Vδ2^+^ TCR repertoires, these TCR repertoires are extremely diverse, non-overlapping, and therefore private [33,34]. In comparison to CMV infections, where an adaptive-like expansion of γδ T cells is evident, the γδ T cell response in HIV-infected individuals seems more complex [82]. Early during HIV infection, Vδ1^+^ T cell expansions are observed, leading to an inverted ratio of Vδ2^+^/Vδ1^+^ cells [82,83]. Expanded Vδ1^+^ cells have been speculated to contribute to the control of HIV replication at mucosal sites of entry [82]. Early spectratyping analysis revealed the polyclonal nature of Vδ1^+^ T cell expansions, as no skewing towards specific TRDV1 or TRGV sequences was observed [84,85], and activation and expansion was associated with microbial translocation in the gut of SIV-infected rhesus macaques [86]. In addition to Vδ1^+^ T cell expansion, Vδ2^+^ T cells are depleted from the circulation correlating with CD4^+^ T cell counts and a loss of pAg reactivity occurs, most likely because specifically Vδ2^+^ cells with Vγ9JP chains disappear, and public clonotypes are lost [87,88]. The depletion of Vδ2^+^ T cells is considered a possible immune evasion strategy of the virus. One mechanism of depletion could be that HIV envelope-mediated cell death by CCR5 is possible in Vδ2^+^ T cells and not Vδ1^+^ T cells [89]. Interestingly, (partial) reconstitution of Vδ2^+^ T cell functionality and Vγ9JP^+^ public clonotypes, including germline-encoded clonotypes, occurs during antiretroviral therapy and is speculated to be mediated by thymic output of Vγ9Vδ2^+^ cells [88]. In the case of Influenza A virus infections, Vγ9Vδ2^+^ T cells have been reported to kill infected cells and provide a major source of IFN-γ [90,91]. Recently, synapse formation and direct killing of H1N1/PR8-infected cells by γδ T cells have been shown ex vivo and Vγ9Vδ2^+^ T cells were described to be the main population of INF-γ producing γδ T cells [66,92].

### 5.2. Bacterial Infections

Mycobacteria are a rich source of bacterial phosphoantigens [13], and thus, phosphoantigen-reactive Vγ9Vδ2^+^ T cells have been implied in protective γδ T cell responses to *Mycobacterium tuberculosis* infections [77,93]. Expansions of Vγ9Vδ2^+^ T cells in pulmonary tuberculosis (TB) have been reported [94,95,96]. Yet, in other reports, comparable to HIV infections, a loss of Vγ9Vδ2^+^ T cells in the blood has been observed in active TB and correlated with disease severity [92,97,98]. Recently, in lungs from HIV-negative patients with active TB, a dominance of Vδ1 (and Vδ3) usage, a bias for Jδ1 and clonal expansions have been shown and most δ-chains were non-overlapping when lungs and blood samples of the same donor were compared [92]. Moreover, highly localized expansions of Vδ1^+^ clonotypes and heterogeneity within individual lung tissues sections in the same study suggested a lung-resident non-recirculating γδ T cell population.

### 5.3. Parasitic Infections

Upon infection with *Plasmodium species*, proliferation and phenotypic changes of γδ T cells have been observed and extensively reviewed elsewhere [99,100,101,102]. Importantly, Vγ9Vδ2^+^ T cells that recognize pAgs produced by the parasite as well as non-Vγ9Vδ2^+^ T cells seem to be implied in the γδ T cells response during malaria [103,104]. For Vγ9Vδ2^+^ T cells, that can be directly cytotoxic for blood stage parasites [104], repertoire studies showed a decrease in JP usage and the occurrence of germline Vγ9JP^+^ clones in neonates as an effect of placental malaria [105,106]. Recently, an oligoclonal expansion of Vδ1^+^ T cells expressing CD38 and PD-1 and lacking the expression of CD27 and CD57, indicative of an early effector phenotype, has been reported in controlled infections of Tanzanian volunteers with *Plasmodium falciparum*, however, as longitudinal data is lacking, it is unclear whether those observations are a result of infection [103].

## 6. Tissue-Resident γδ TCR Repertoires

The V-gene usage of γδ T cells is not only associated with their function and ontogeny, but also with their tissue localization. While pAg-inducible Vγ9Vδ2^+^ T cells are predominant in peripheral blood lymphocytes, the majority of γδ T cells localizing to solid organs or mucosal tissues often express non-Vγ9Vδ2^+^ TCRs. This phenomenon of V-gene usage linked to tissue distribution seems highly similar to their mouse counterparts (reviewed in [23]). However, knowledge of the human γδ TCR repertoire composition in healthy and diseased tissues remains fragmented. There is evidence that γδ TCR repertoires have an oligoclonal distribution in healthy liver, spleen, lymph node and lung [66,107]. An NGS analysis of intrahepatic Vδ2^–^ γδ T cells showed that CD69^+^ tissue-resident Vδ2^–^ T cells are characterized by unique TCR clones, while expanded TCR clones of liver-infiltrating Vδ2^–^ T cells are present in the blood of the respective donor, albeit with lower abundance. Interestingly, the recruitment of adaptive-like Vγ9^–^Vδ2^+^ cell subsets to the liver seems evident [31]. In healthy individuals, lung γδ T cells are enriched for Vδ2^+^ T cells, while patients with active tuberculosis were reported to have elevated Vδ1^+^ T cell numbers that display skewed TCR repertoires. However, there was no dominant Vδ1^+^ γδ TCR clone or motif among patients with active tuberculosis. The distribution of TCR repertoires proposed that presumably lung-infiltrating Vδ1^+^ T cells underwent an adaptive-like clonal expansion during active tuberculosis [92]. In contrast, tissue-resident NKG2D^+^ and CD69^+^ Vδ1^+^ T cells were reported to have oligoclonal TCR repertoires in healthy breast tissues that remained stable after tumor-infiltration, consistent with their innate-like features such as NKG2D-driven activation [108,109].

Similar innate-like phenotypes can be ascribed to human gut-resident Vγ4^+^ intestinal epithelial γδ T cells (IELs), that are CD69^+^ and express the natural cytotoxicity receptors Nkp46 and/or Nkp44 [38,110]. Human Vγ4^+^ IELs are shaped and selected by the BTNL-like molecules, BTNL3 and BTNL8, that are exclusively expressed in the human gut epithelial [14]. TCR repertoire analysis of total or Nkp46^+^ γδ IELs isolated from healthy tissues gave evidence for a relatively clonal TRG and TRD repertoire, enriched for public Vγ4^+^ and private Vδ1^+^ T cell clones [38,39,110]. Importantly, celiac disease leads to a loss of the BTNL-induced Vγ4^+^ IEL compartment, that cannot be restored after gluten-free diet [38]. Notably, an innovative approach of single-cell TCR repertoire analysis of γδ IELs of celiac disease patients noticed a higher TCR repertoire diversity, due to loss of innate-like Vγ4^+^ IELs, but did not identify public γδ T cell clones that may recognize defined disease-associated ligands [39]. Moreover, in colorectal cancer patients with higher numbers of innate-like cytotoxic Nkp46^+^ Vγ4Vδ1^+^ T cells were correlated with a better clinical outcome [110]. Altogether, TCR-seq analysis can be a valuable method to distinguish tissue-resident and circulating γδ T cell clones and characterize adaptive-like versus innate-like expansions of tissue γδ T cells in healthy and diseased individuals.

## 7. Conclusions

γδ T cells have long been understood as unconventional innate-like T cells with TCRs of only limited diversity, predisposed for rapid recognition of highly conserved antigens. Now, a more nuanced view of γδ T cell function is emerging. While Vγ9Vδ2^+^ T cells are still viewed as largely invariant innate-like T cells, current research focusses on the question whether some versions of the Vγ9Vδ2^+^ TCR cells are superior to others. Future studies combining single cell TCR and total RNA sequencing should be very instructive. Regarding non-Vγ9Vδ2^+^ T cells, it is clear that they play a more sophisticated role and can establish a hitherto unrecognized form of individual adaptive immune surveillance [33,34,71,107]. Clonotype-specific expansions of non-Vγ9Vδ2^+^ T cells occur in multiple diseases, but are also observed in healthy individuals. This has led to the hypothesis that the non-Vγ9Vδ2^+^ TCR repertoire could serve as a log-file, reflecting the immunological history of individual antigen challenges.

## Figures and Tables

**Figure 1 cells-09-00800-f001:**
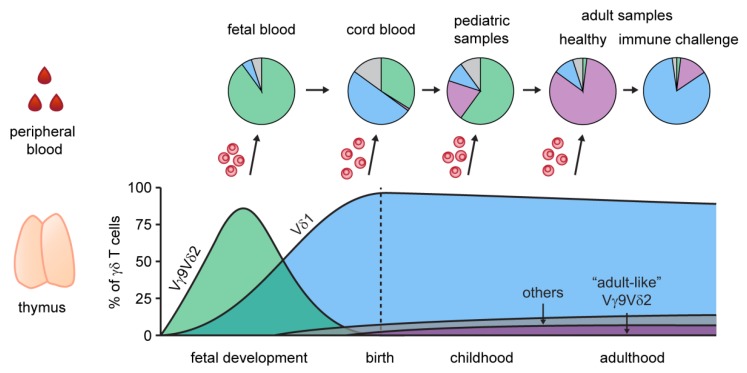
Developmental waves of human γδ T cells. The γδ T cell population in the human thymus shows characteristic waves of γδ T cell subpopulations, with distinct V-gene usage (lower panel). Schematic proportions of Vγ9Vδ2^+^ (fetal: green, adult-like: purple), Vδ1 (blue) and γδ T cells using other V-genes (“others”: gray) are shown as a percentage of all thymic γδ T cells. Migration of thymic γδ T cells and extrathymic changes subsequently contribute to the formation of the adult peripheral blood γδ T cell compartment (upper panel). The composition of peripheral blood γδ T cells from second trimester fetal blood to adults is illustrated by pie charts. Typical clonal expansions of Vδ1^+^ T cells were observed in a multitude of immune challenges (e.g., CMV), and indicate an adaptive-like γδ T cell response.

**Table 1 cells-09-00800-t001:** Human γδ T cell subsets by high-throughput TCR sequencing. Table summarizes features of γ-chain (TRG) and δ-chain (TRD) repertoires of the major Vγ9Vδ2^+^ (fetal and adult-like) and non-Vγ9Vδ2^+^ T cell subsets (Vγ9^-^Vδ2^+^, Vδ1^+^, Vδ3^+^, Vγ8Vδ1/2^+^ and Vγ4^+^). Nomenclature of gene segments according to Lefranc/Forster [25], in brackets according to IMGT [26] and Strauss et al. [27].

Human γδ Subsets	TRG Repertoire	TRD Repertoire	Characteristics
Fetal Vγ9Vδ2^+^ [28]	- Semi-invariant Vγ9JP (IMGT: TRGV9/TRGJP, Strauss: Vγ2/Jγ1.2) - Shared CDR3γ sequences and length homogenization - Germline-encoded clonotypes: Short-homology repeats	- Predominant Vδ2Jδ3 usage (IMGT: TRDV2/TRDJ3) - Private and shared CDR3δ - Shorter CDR3δ lengths	- Phosphoantigen-reactive γδ T cell subset - Polyclonal expansion upon antigen stimulation - Extrathymic, postnatal expansion
Adult-like Vγ9Vδ2^+^ [28,29,30]	- Semi-invariant Vγ9JP - Shared CDR3γ sequences and length homogenization - Germline-encoded clonotypes: N additions	- Predominant Vδ2Jδ1 usage (IMGT: TRDV2/TRDJ1) - Private CDR3δ	- Phosphoantigen-reactive γδ T cell subset - Originate from postnatal thymus - Polyclonal expansion upon antigen stimulation - Extrathymic expansion
Vγ9^–^Vδ2^+^ [31,32]	- Diverse Vγ chains - Private CDR3γ	- Private CDR3δ	- Clonal expansion in CMV - Liver infiltrating/tissue homing
Vδ1^+^ [33,34]	- Diverse Vγ chains - Private CDR3γ	- Vδ1 usage (IMGT: TRDV1) - Private CDR3δ	- Clonal expansion in CMV
Vδ3^+^ [34,35]	- Diverse Vγ chains?	- Vδ3 usage (IMGT: TRDV3) - Clonal/oligoclonal repertoire? - Private CDR3δ	- Clonal expansion in some HCV patients - Moderate clonal focusing
Fetal Vγ8Vδ1/2^+^ [36,37]	- Vγ8JP1 usage (IMGT: TRGV8/TRGJP1, Strauss: Vγ1.8/Jγ1.1) - Public, short CDR3γ - Germline-encoded clonotypes: Short-homology repeats	- Vδ1 or Vδ2 usage - Public, short CDR3δ - Germline-encoded clonotypes: Short-homology repeats	- CMV-responsive in utero - Lin 28b-driven intrinsic priming for IFN-γ and granzyme expression - Invariant/public clonotypes - Not present in adults
Vγ4^+^ [38,39]	- Public Vγ4 chains (IMGT: TRGV4, Strauss: Vγ1.4) - Clonal repertoire	- Private Vδ1 chains - Clonal repertoire	- Intestinal epithelial γδ T cell subset - Innate-like phenotype - Loss of Vγ4+ in celiac disease

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
