# Peer review of "Human γδ TCR Repertoires in Health and Disease"

_cells, 2020, doi:10.3390/cells9040800_

Round 1

Reviewer 1 Report

The review provides a comprehensive overview of the field of human gdT cells both in physiological conditions as well as in bacterial parasitic and viral infections. The characteristics of the subpopulations of gd T cells in the thymus and peripheral blood during ontogeny is clearly summarized in figure 1. It is also highlighted the role of tissue resident gd T cells during infection. The author's group has given with its work a substantial contribution to the study of gd T cells and here it is also extensively summarized the literature data on the topic.
I do not see any relevant points to indicate except for few typos throughout the text.

Author Response

Response to Reviewer 1 Comments

The review provides a comprehensive overview of the field of human gdT cells both in physiological conditions as well as in bacterial parasitic and viral infections. The characteristics of the subpopulations of gd T cells in the thymus and peripheral blood during ontogeny is clearly summarized in figure 1. It is also highlighted the role of tissue resident gd T cells during infection. The author's group has given with its work a substantial contribution to the study of gd T cells and here it is also extensively summarized the literature data on the topic.

I do not see any relevant points to indicate except for few typos throughout the text.

Response: We thank the reviewer for the positive recommendation of our manuscript; we revised the text accordingly and performed a thorough check of spelling and grammar. We also added a new table as suggested by Reviewer 2 showing an overview of the main human γδ T cell subsets

Reviewer 2 Report

Fichtner et al provide a comprehensive overview of several recent articles that include TCR repertoire data and a detailed background to appreciate these studies. The health side of the article is well documented, however, the disease section is surprisingly limited due to the lack of studies addressing this area. Comments: Abstract: "While the theoretical TCR repertoire diversity of γδ T cells is estimated to exceed the diversity of αβ T cells by far, γδ T cells are still understood as more invariant T cells that only use a limited set of γδ TCRs and function between innate and adaptive immunity." This is a confusing statement and very long. Break this down, I understand your point but ending in "functioning between innate and adaptive immunity" without any context is confusing. "namely Vδ2+ and Vδ1+ , also known as Vδ2– T cells". This doesn't make sense? "Of those two subsets, the Vδ2+ T cells seem to better fit into a role of innate T cells with invariant TCR usage as compared to the more adaptive Vδ1+ γδ T cells". But you just mentioned three subsets (VD1, VD2 and VD2-)? VD2 have semi-invariant TCR usage. Are VD1s adaptive - NKG2D responses, Vg4 restricted BNTL3 interactions? - Mayassi et al. Cell. 2019, Wu et al STM 2019 and Willcox et al Immunity 2019. Intro: "detected at surprisingly variable frequencies" - across several cohort studies in blood gd are very consistently 5-10% of total T cells. What do you mean here? Could mention the TRA names for TRDV4,, V8 etc. Not all Vd2 TCRs pair with Vg9 and this population is prominent at birth (within VD2+ cells) and in CMV infection - Davey et al Nat Comm 2018. Should discuss the authors work on BTN2A1 and the interpretation of BTN3A1's role in PAg sensing - i.e. probably not a TCR ligand - in contrast to Rigau et al Science 2020's conclusion. CMV repertoires probably should include Vd9-VD2+ T cells as mentioned above and also noted as a tissue resident population. A table to define the main subsets of human gd T cells would be useful including the names they can be called - i.e. non-Vg9Vd2, VD2-, VD1+, VD2+ etc. Just for clarity. "Recently, an oligoclonal expansion of Vδ1+ T cells expressing CD38 and PD-1 and lacking expression of CD27 and CD57, indicative of an early effector phenotype, has been reported in controlled infections of Tanzanian volunteers with Plasmodium falciparum." This study is confusing, and only reports repertoires at a fixed timepoint (not longitudinally), also while up-regulation of CD38 and PD-1 are evident - numerical expansion is no. Without baseline date it's unclear if these oligoclonal repertoires are as a result of Plasmodium or simply the underlying clonotypes before challenge. Please check the text as several mistakes and missing words are evident.

Author Response

Response to Reviewer 2 Comments

Fichtner et al provide a comprehensive overview of several recent articles that include TCR repertoire data and a detailed background to appreciate these studies. The health side of the article is well documented, however, the disease section is surprisingly limited due to the lack of studies addressing this area.

Comments:

  1. "While the theoretical TCR repertoire diversity of γδ T cells is estimated to exceed the diversity of αβ T cells by far, γδ T cells are still understood as more invariant T cells that only use a limited set of γδ TCRs and function between innate and adaptive immunity." This is a confusing statement and very long. Break this down, I understand your point but ending in "functioning between innate and adaptive immunity" without any context is confusing. "namely Vδ2+ and Vδ1+ , also known as Vδ2– T cells". This doesn't make sense? "Of those two subsets, the Vδ2+ T cells seem to better fit into a role of innate T cells with invariant TCR usage as compared to the more adaptive Vδ1+ γδ T cells". But you just mentioned three subsets (VD1, VD2 and VD2-)? VD2 have semi-invariant TCR usage. Are VD1s adaptive - NKG2D responses, Vg4 restricted BNTL3 interactions? - Mayassi et al. Cell. 2019, Wu et al STM 2019 and Willcox et al Immunity 2019.

Response: We thank the reviewer for this suggestion to clarify the abstract and changed the text accordingly to not create any confusion (p. 1, line 11 and line 13-20).

  1. Intro: "detected at surprisingly variable frequencies" - across several cohort studies in blood gd are very consistently 5-10% of total T cells. What do you mean here?

Response: We thank the reviewer for pointing out the confusing phrasing and changed the sentence on page 1, line 26: “detected at frequencies of 5-10% of T cells in the peripheral blood of human adults”

  1. Could mention the TRA names for TRDV4,, V8 etc.

Response: We now mention the TRA names for the V4-8 gene segments on p.2, line 85-86: “(TRDV1-3, TRAV14/DV4, TRAV29/DV5, TRAV23/DV6, TRAV36/DV7 and TRAV38/DV8)”

  1. Not all Vd2 TCRs pair with Vg9 and this population is prominent at birth (within VD2+ cells) and in CMV infection - Davey et al Nat Comm 2018.

Response: Thank you for this comment, we now try to depict this subset more clearly in the text (see page 4 line 109, p. 7 line 263 and p. 8 line 323). We also mention this subset in the newly added table.

  1. Should discuss the authors work on BTN2A1 and the interpretation of BTN3A1's role in PAg sensing - i.e. probably not a TCR ligand - in contrast to Rigau et al Science 2020's conclusion.

Response: Thank you for this suggestion, due to the recently published manuscript of Karunakaran et al. we are now able to discuss Vγ9Vδ2-BTN2A1 interactions and pAg sensing taking both publications into account (p. 3 line 96  starting from: “BTN2A1 has been shown…”)

  1. CMV repertoires probably should include Vd9-VD2+ T cells as mentioned above and also noted as a tissue resident population.

Response: We now included Vγ9-Vδ2+ T cells in the CMV repertoire sections (page 7) and in the part about tissue-resident populations (page 8), see comment 4

  1. A table to define the main subsets of human gd T cells would be useful including the names they can be called - i.e. non-Vg9Vd2, VD2-, VD1+, VD2+ etc.

Response: We thank the reviewer for this great suggestion. We created a table of human γδ T cell subsets including comments on TRG and TRD repertoire and some general characteristics (page 3)

  1. Just for clarity. "Recently, an oligoclonal expansion of Vδ1+ T cells expressing CD38 and PD-1 and lacking expression of CD27 and CD57, indicative of an early effector phenotype, has been reported in controlled infections of Tanzanian volunteers with Plasmodium falciparum." This study is confusing, and only reports repertoires at a fixed timepoint (not longitudinally), also while up-regulation of CD38 and PD-1 are evident - numerical expansion is no. Without baseline date it's unclear if these oligoclonal repertoires are as a result of Plasmodium or simply the underlying clonotypes before challenge.

Response: We thank the reviewer for this assessment, we changed the sentence on p.8 line 310 to “Recently, an oligoclonal expansion of Vδ1+ T cells expressing CD38 and PD-1 and lacking expression of CD27 and CD57, indicative of an early effector phenotype, has been reported in controlled infections of Tanzanian volunteers with Plasmodium falciparum, however, as longitudinal data is lacking, it is unclear whether those observations are a result of infection[1].”

  1. Please check the text as several mistakes and missing words are evident.

Response: We thoroughly checked our manuscript to eliminate spelling errors and other mistakes.